# Learning Time-dependent PDE Solver using (Recurrent) Message Passing Graph Neural Networks

## Abstract

One of the main challenges in solving time-dependent partial differential equations is to develop computationally efficient solvers that are accurate and stable. Here, we introduce a graph neural network approach to finding efficient PDE solvers through learning using message-passing models. We first introduce domain invariant features for PDE-data inspired by classical PDE solvers for an efficient physical representation. Next, we use graphs to represent PDE-data on an unstructured mesh and show that message passing graph neural networks (MPGNN) can parameterize governing equations, and as a result, efficiently learn accurate solver schemes for linear/nonlinear PDEs. We further show that the solvers are independent of the initial trained geometry, i.e. the trained solver can find PDE solution on different complex domains. Lastly, we show that a recurrent graph neural network approach can find a temporal sequence of solutions to a PDE.

## 1 Introduction

Physical phenomena are generally modeled through partial differential equations (PDEs) that govern the dynamic evolution or static solution of a physical system. Numerically solving partial differential equations is an important aspect of scientific and mathematical modeling in a broad range of fields including physics, biology, material science, and finance. There have been many efforts to develop efficient and accurate numerical solvers for PDEs using different techniques including finite difference (LeVeque, 2007; Shashkov & Steinberg, 2018), finite volume (Eymard et al., 2000; Brenner et al., 2008), and finite element schemes (Reddy, 2014; Wriggers, 2008). While these methods have been successful in producing accurate solutions, major challenges for accelerating and reducing computational cost when the governing PDE is known, and also determining the governing PDE when the physical system is unknown, remains to be addressed, problems such as those in climate modeling, turbulent flow, contact problems, or plastic material deformation. With the recent developments in deep learning, faster algorithms have been proposed to evaluate the response of a physical system, using only observational data. While deep learning approaches, such as multi-layer perceptron (MLP) or convolutional neural networks (CNNs), are powerful in learning PDE solutions, they are, however, restricted to a specific discretization of the physical domain in which they are trained. As a result, the learned model is limited to a specific domain and can not be generalized to solve on different domains or for different discretizations, although the underlying physics remains to be the same. New training is required for any change in the physical domain or discretization. Here we propose a discretization and domain invariant neural network time-dependent PDE solver based on message passing graph neural nets (MPGNN) which is trained on a sample domain with different discretizations. The trained MPGNN can then be used to solve for different discretization or even on other domains as long as the underlying physics remains the same. We further show that a recurrent version of MPGNN can be used to find a temporal sequence of solutions to a PDE.

## 2 Related Works

One class of neural-net-based PDE solvers focuses on using neural networks as proxies of PDEs and aims at finding the solution by minimizing a loss that corresponds to the solution satisfying the governing equations and the boundary conditions (Raissi et al., 2017a;b; Lagaris et al., 1998; Weinan

& Yu, 2018; Sirignano & Spiliopoulos, 2018; Khoo et al., 2021). Although such an approach helps to find the one-time solution of a PDE with an instance of parameters, a slight modification to the PDE parameters, boundary conditions, or the domain requires re-training of the network.

Another approach to solving PDEs is to use convolutional neural nets and snapshots of observations over the discretized input domain and to learn the dynamic evolution of a PDE (Long et al., 2018; Shi et al., 2020). Further modifications such as using residual connections (Ruthotto & Haber, 2020), or autoregressive dense encoder-decoder (Geneva & Zabaras, 2020), or symbolic multi-layer neural network (Long et al., 2019) in addition to the CNN can be used to improve the results. While these models do not require prior knowledge of the PDE, they are limited to domain discretization (as a result cannot be generalized to arbitrary domains) and are limited to certain time discretization (as a result unable to handle temporally and spatially sparse or non-uniform observations).

Inspired by the discretization techniques in solving PDEs, a class of methods uses observational data to learn the discretization approximation required for the updates in classical computational PDE solver methods (Bar-Sinai et al., 2019; Kochkov et al., 2021; Zhuang et al., 2021; Han et al., 2018). In this approach, a neural network is used for better interpolation at coarse scale to be used in the framework of traditional numerical discretization. These methods are used in conjunction with classical numerical methods and can improve the accuracy and accelerate the solutions of the traditional numerical schemes (Kochkov et al., 2021). Although these methods have been shown to generalize to new parameter regimes of a PDE, they are still bounded to the initially trained discretization and can not be used for arbitrary domains without re-training.

Lastly, a class of neural PDE solvers focus on graph representation of the discretized mesh data-structure to approximate the PDE solution (Li et al., 2020a;b; Iakovlev et al., 2020; Belbute-Peres et al., 2020). The numerical solution of a PDE is an approximation of the solution on discrete locations comprising a discretized mesh of continuous space. Each node represents a region in the continuous space and the approximate solution of the PDE in that region is assigned to the representative node. The discretized mesh forms a graph where each node is used to model the state of the system and forms a connectivity graph connecting to the neighboring nodes. This method has successfully been used to solve time-independent PDEs with different mesh sizes on the same physical domain (Li et al., 2020a). The connectivity and the location of the nodes can further be optimized to learn the solution with different levels of precision (Alet et al., 2019). If the PDE includes long-range interactions, which happens mostly in time-independent PDEs, a multi-level graph neural network framework to encapsulate long-range interactions can be used to improve the results (Li et al., 2020b). In contrast to time-independent PDEs, in the case of time-dependent PDEs, it has been shown that a continuous-time model similar to physics informed neural nets but with a graph neural network can be used to recover system's dynamics with sparse observational data recorded at irregular times (Iakovlev et al., 2020; Poli et al., 2019). Recently, it have been shown that message passing graph neural networks can be used to implement powerful physical simulation engines (Pfaff et al., 2020; Sanchez-Gonzalez et al., 2020). The state of a physical system can be expressed using a particle-based method as a reduced order model. The particles are then expressed as nodes in a graph and the message passing neural network learns to compute the dynamics of the particles(Sanchez-Gonzalez et al., 2020). In addition to particle-based methods, mesh-based methods have been shown to be successful in physical simulations (Pfaff et al., 2020). Such graph-based models, first encodes the input data into a latent space and then process it in the latent space (reduced model), and to obtain the physical results decode the data back to the physical space. Here, we first show why graph neural networks can generalize to learn fast PDE solvers inspired by finite difference schemes. We introduce domain invariant features and boundary conditions inspired by classical PDE solvers to improve the generalization of the learned PDE solver operator. With the introduced features, we show that message passing graph neural network architecture efficiently fits the classical PDE solvers and can learn time-stepping/solver operators for linear and nonlinear PDEs with different boundary conditions. We further demonstrate that our trained graph neural network solver can be generalized to solve PDEs on physical domains different from the domain that it is trained on. This is beneficial to train GNN on a sample of small domains for even with unknown dynamics, and further, explore the dynamic behavior on different larger physical domains. Lastly, we show that a recurrent version of our MPGNN can be used to predict the temporal sequence of solutions to a PDE.

## 3 TIME-DEPENDENT PDEs

We consider continuous dynamical system $u(\mathbf{x}, t) \in \mathbb{R}$ evolving over time $t \in \mathbb{R}^+$ and spatial coordinate $\mathbf{x} \in \Omega \subset \mathbb{R}^d$ where $\Omega$ is a bounded $d$-dimensional domain. We assume the system is governed by a partial differential equation of the form

$$u_t = \mathcal{N}[u; \boldsymbol{\lambda}] \tag{1}$$

where $\mathcal{N}[\cdot; \boldsymbol{\lambda}]$ denotes the linear/nonlinear differential operator(s) parameterized by the vector $\boldsymbol{\lambda}$. In the above general form, the temporal evolution of the current state $u_t$ depends on the differential operator $\mathcal{N}[\cdot; \boldsymbol{\lambda}]$ which may include various spatial derivatives of the state including $\nabla u, \nabla^2 u$, etc. Depending on the differential operator $\mathcal{N}$, appropriate boundary conditions on $\partial\Omega$ is required for a well-posed PDE with a unique solution. Such PDE model is the cornerstone of mathematical models and is widely used to model various systems, from fluid dynamics, thermal sciences, to acoustics, and quantum mechanics. As an example, $u_t = \mathcal{N}[u; \boldsymbol{\lambda}] = \lambda_1 \nabla^2 u + \boldsymbol{\lambda_2} \cdot \nabla u$ constitutes a convection diffusion equation for $u \in \mathbb{R}$ as variable of interest, where $\boldsymbol{\lambda} = \lambda_1, \boldsymbol{\lambda_2}$ are the diffusitivity and the velocity field vector with which the quantity is moving with.

The state of the system at each time can be obtained using its initial state and time integration as $u(\mathbf{x}, t) = u(\mathbf{x}, 0) + \int_0^t \mathcal{N}(u; \boldsymbol{\lambda}) dt$. Numerous numerical techniques such as Finite Elements, Spectral Methods, Finite Difference, or Finite Volume techniques have been developed to efficiently approximate the differential operator $\mathcal{N}(\cdot; \boldsymbol{\lambda})$ and solve for a dynamical system over time. In all numerical schemes, the domain is first discretized with a mesh, the differential operator is approximated locally using neighboring points, and the solution is calculated over small time steps using a time integrator such as Euler's scheme, i.e.,

$$u^{n+1}(\mathbf{x}_i) = u^n(\mathbf{x}_i) + \delta t \mathcal{F}(u^n(\mathbf{x}_i), \nabla u^n(\mathbf{x}_i), \nabla^2 u^n(\mathbf{x}_i), \cdots; \boldsymbol{\lambda}(\mathbf{x}_i)) \tag{2}$$

where the superscript $n$ shows the solution over discretised time $t_n$, and the differential operator $\mathcal{N}(u; \boldsymbol{\lambda}) = \mathcal{F}(u, \nabla u, \nabla^2 u, \cdots; \boldsymbol{\lambda})$ shows that it contains information about local spatial derivatives. As an example, consider solving heat equation, $u_t = D\nabla^2 u$, where $D$ is the diffusion constant and $\nabla^2 u = \partial^2 u/\partial x^2 + \partial^2 u/\partial y^2$, on an structured grid shown in Fig. 1a. Let $u_{i,j}^n$ be the discretized solution at time $t = n\delta t$ and spatial location $x = i\delta x$ and $y = j\delta y$ where $\delta t, \delta x,$ and $\delta y$ are the time, horizontal, and vertical spatial discretization respectively. The time and spatial derivatives in the heat equation can be expanded using Taylor series at each discretized point, where $\partial u_{i,j}^n/\partial t = (u_{i,j}^{n+1} - u_{i,j}^n)/\delta t$, $\partial^2 u_{i,j}^n/\partial x^2 = \left(u_{i+1,j}^n - 2u_{i,j}^n + u_{i+1,j}^n\right)/\delta x^2$,

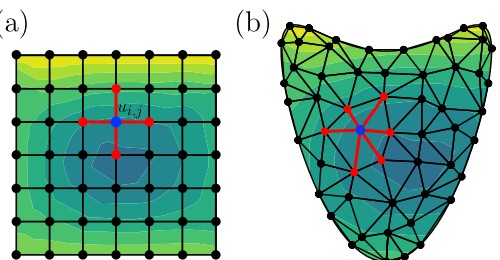

Figure 1: (a) A square domain with structured mesh discritization (b) A curved domain with an unstructured

and etc. Re-writing the equation for an arbitrary discretized point, we find $u_{i,j}^{n+1} = u_{i,j}^n + \delta t \mathcal{F}$ where $\mathcal{F} = \alpha\left(u_{i,j+1} - u_{i,j}\right) + \alpha\left(u_{i,j-1} - u_{i,j}\right) + \beta\left(u_{i+1,j} - u_{i,j}\right) + \beta\left(u_{i-1,j} - u_{i,j}\right)$, where $\alpha = D\delta t/\delta x^2$ and $\beta = D\delta t/\delta y^2$. Solving for this equation for all the points along with the boundary conditions the updates for the discretized points can be achieved. Note that here the update rule can be seen as the summation of updates that only depend on neighboring points. Although in this example with a simple linear equation and a structured grid it was easy to find the update rule $\mathcal{F}$, given an arbitrary domain that requires a triangular mesh for discretization (see Fig. 1b) and a nonlinear governing equation the update rule is not straight forward to be worked out. Our objective here is to learn the approximation of the differential operator using a graph representation of the domain with message passing neural networks. Since in a general PDE the differential operator $\mathcal{F}(u, \nabla u, \nabla^2 u, \cdots; \boldsymbol{\lambda})$ contains local spatial derivatives and only local neighboring values at a point are relevant to approximate the differential operator at that point. As a result, a graph neural network is a promising framework to approximate the above right-hand side for the next value predictions.

## 4 CONTRIBUTIONS

In this paper, we propose a graph-based model to learn domain-invariant and free-form solvers for a PDE on arbitrary spatial discretization using message passing neural networks. Our method here is inspired by the finite difference method and the possibility of approximating a partial differential equation using discrete point stencils. Here, we use graph neural networks as a nonlinear function approximator and use the simulated data to learn the required stable stencils. In order to show that that the graph neural network has learned the correct PDE solvers, we test our learned graph network to find PDE solutions in a domain with a different geometry and mesh discretization. We find that the trained model on a sample domain can be used to predict the PDE solution on other physical domains and mesh discretization. This is only possible by creating relevant features inspired by classical PDE solver techniques to represent the differential operator. Our contributions are: (i) introducing locally invariant feature representation inspired by classical PDE solvers to efficiently learn a differential operator solver; (ii) showing graph representation with message passing neural networks that can parametrize PDE solvers with a domain-invariant representation; (iii) obtaining robust high-performance models for various linear/nonlinear PDEs with arbitrary spatial discretization and showing that it can generalize to arbitrary physical domains, (iv) proposing a recurrent message passing graph neural network approach to predict a temporal sequence of PDE solution over time.

## 5 GRAPH NEURAL NETWORKS FOR PDES

Let $\mathcal{G} = (\mathcal{V}, \mathcal{E})$ be graph representation of a mesh with nodes $\mathcal{V} = \{\mathbf{x}_i\}_{i=1}^N$ where $\mathbf{x}_i$ denotes the positions, and edges $\mathcal{E} = \{e_{ij}\}$ where $e_{ij}$ represents the connecting neighboring points at $\mathbf{x}_i$ and $\mathbf{x}_j$. Given a physical domain, we use a uniform random distribution of points for the nodes, and use Delaunay triangulation to find the neighboring points. We denote neighbors of node $i$ with $\mathcal{M}(i) = \{j | e_{ij} \in \mathcal{E}\}$. We further assign the node and edge attributes with $\mathbf{u}_i$ and $\mathbf{e}_{ij}$ respectively (see Fig. 2). In a message-passing neural network (Gilmer et al., 2017), we propagate the latent state of nodes $\mathbf{u}_i$ of for $K$ layers, where at each layer $k$, we have

$$\mathbf{u}_i^{(k)} = \gamma^{(k)} \left( \mathbf{u}_i^{(k-1)}, \frac{1}{|\mathcal{M}(i)|} \sum_{j \in \mathcal{M}(i)} \phi^{(k)} \left( \mathbf{u}_i^{(k-1)}, \mathbf{u}_j^{(k-1)}, \mathbf{e}_{ij} \right) \right) \tag{3}$$

where $\phi^{(k)}$ and $\gamma^{(k)}$ are differentiable deep neural networks. Note that $N = |\mathcal{M}(i)|$ represents the number of neighbors for node $i$, and furthermore instead of the average used in equation 3, other permutation invariant aggregation function such as sum or max can also be used. Since equation (3) is an approximation of equation (2) where $\mathcal{N}(u; \lambda)$ includes various spatial differentials, we take our edge features to include $\mathbf{x}_j - \mathbf{x}_i$ for $j \in \mathcal{M}(i)$, and also $\boldsymbol{\lambda}(\mathbf{x}_{ij})$ which is the PDE parameters at the midpoint of the edge $\mathbf{x}_{ij} = (\mathbf{x}_i + \mathbf{x}_j)/2$. We further can include, higher derivatives of the PDE parameters such as $\nabla\boldsymbol{\lambda}(\mathbf{x}_{ij}), \nabla^2\boldsymbol{\lambda}(\mathbf{x}_{ij})$ for a better approximation representation. At each point in time, we set $\mathbf{u}_i = [u(\mathbf{x_i}, t), u(\mathbf{x_i}, t - \delta t), \cdots, u(\mathbf{x_i}, t - n\delta t)]$ as the last $n$ snapshots for the initial latent space and use it to create the desired $\mathbf{u}_i^{(K)} = [u(\mathbf{x_i}, T), u(\mathbf{x_i}, T - \delta t'), \cdots, u(\mathbf{x_i}, T - m\delta t')]$ as the desired last $m$ frames of the solution. The main assumption here is that the derivatives can be approximated using the graph nodes, which is possible in most physical simulations where the solutions are smooth. Additionally, note that the predicted values here include all inside and boundary nodes.

We use multilayer perceptron for the $\gamma^{(k)}$ and $\phi^{(k)}$ with three hidden layers. Note that three hidden layers, allows each point to connect to the third order neighbors (i.e., neighbors of neighbors of neighbors), which potentially only allows for a maximum $6^{th}$ order derivative estimation. In general, the node features consist of the solution in previous time steps and edge features, motivated by the logic of solving PDEs, are made up of the distance between connecting neighboring nodes, along with the PDE parameters calculated at the center of the edge. We also add an extra feature to the node attributes showing that if the node lies on the boundary or not. This extra feature is 1 if the node lies on the boundary and 0 otherwise. Our decision on the node and edge features might slightly differ for different equations, and we point out if there is any change in the features set.

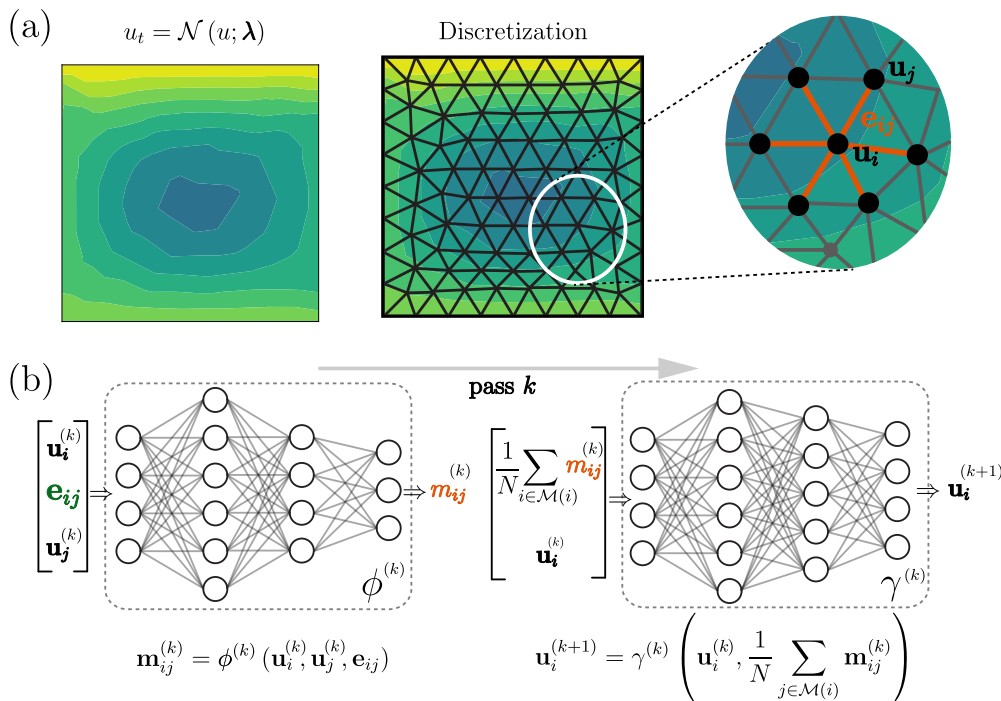

Figure 2: (a) Example of a physical domain governed by the time dependent PDE $u_t = \mathcal{N}(u; \boldsymbol{\lambda})$, and the spatial discretization that represents a graph structure. The inset figure shows the node features $\mathbf{u}_i$ and the edge features on the mesh $\mathbf{e}_{ij}$. (b) Graphical representation for one layer pass of message passing neural network: given node values $\mathbf{u}_i^{(k)}$ and edge attributes $\mathbf{e}_{ij}$, a message is generated for each edge such that $\mathbf{m}_{ij}^{(k)} = \phi^{(k)}\left(\mathbf{u}_i^{(k)}, \mathbf{u}_j^{(k)}, \mathbf{e}_{ij}\right)$ where $\phi^{(k)}$ is a neural network; next, given average message value $\frac{1}{N}\sum_{j \in \mathcal{M}(i)} \mathbf{m}_{ij}^{(k)}$ and the node features $\mathbf{u}_i^{(k)}$, the next feature set $\mathbf{u}_i^{(k+1)}$ for the nodes are obtained using another neural network $\gamma^{(k)}$.

## 6 TEST CASES

In this section, we go through different linear/nonlinear PDEs for physical systems and evaluate the performance of our modeling. First, we start with the time-dependent heat equation and show how our model can learn to predict the future state(s) (see section 6.1), and more importantly, the learned model can be used for predictions in new physical domains different from initial learned domain (see figure 4). To predict the sequence of temporal data of a PDE, we use a recurrent message passing graph neural network approach and show that our model is able to predict the sequence of PDE temporal data (see figure 6). Next, to show that the model is able to learn the nonlinearities, we focus on the Navier-Stokes equation to predict future solutions with an arbitrary discretization (see section 6.2). Lastly, to show that the model is able to include PDE parameters, we focus on the advection-diffusion equation and by including the PDE parameters as edge attributes, we show that the model is able to learn to predict for various PDE parameters (see section 6.3).

### 6.1 HEAT EQUATION

Time-dependent heat equation in two dimensions can be written as

$$u_t = \nabla^2 u, \quad \mathbf{x} \in \Omega \tag{4}$$

which can be solved with an initial condition $u(\mathbf{x}, t = 0)$ and the Dirichlet boundary condition, i.e., $u(\mathbf{x} \in \partial\Omega, t) = u_0$, where $\partial\Omega$ denotes domain's boundary. We chose a square grid $[0, 1] \times [0, 1]$, and use Firedrake (Rathgeber et al., 2016) with a characteristic length of $\delta x = 0.0625$ for a

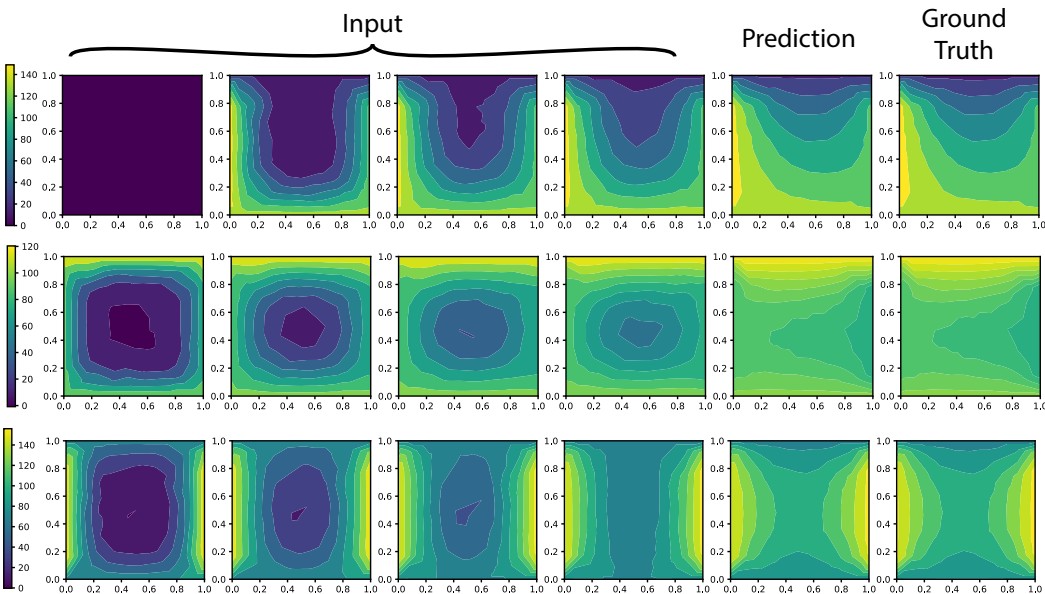

Figure 3: Message passing graph neural network learning to predict the heat equation: Input data frames (first four columns), prediction (fifth column), and ground truth (the last column) for three different test cases in each row. The simulation data corresponds to the heat equation 4 with different boundary conditions. The first four columns correspond to the results with $\Delta t = 20\delta$ time difference, and the network predicts the results for the next frame with $\Delta t = 20\delta t$ after the last frame. The average MSE loss for the test data is $5.1 \times 10^{-6}$.

triangular mesh and time-stepping of $\delta t = $8e-4. Note that the mesh in the simulations are generated using built-in mesh generator in Firedrake. In order to sample the data for training the MPGNN, we construct a Delaunay triangulation of uniformly distributed nodes (Poisson point process) in the domain. We chose different Dirichlet boundary conditions where we set top, left, right, and bottom boundary conditions to different constant values of $u_0 \in [0, 200]$. Note that $u$ for different simulation also remains in the same range as $u_i \in [0, 200]$. For each simulation, we use a newly generated mesh and set of boundary conditions. We set record the data every $20\delta t$, and use four subsequent observations for the input data (i.e., $n = 4$), and predict the next frame (i.e., $m = 1$). As a result, we input frames, $0, 20\delta t, 40\delta t, 60\delta t$ and predict the frame $80\delta t$. Since per each simulation, we can further use the initial frame to be any of 0 to $20\delta t$ frames, we can generate 20 data inputs per simulation. We generate 1000 simulations with different meshes and boundary conditions, and with that, we create 20,000 input data. We define a MSE loss between the network output and the true values as $\sum_i \|\hat{u}_i - u_i\|^2 / N$, where $\hat{u}_i, u_i$ are the network prediction and ground truth for the node $i$ value and $N$ is the number of nodes. We further use three layers for the message passing graph neural network with 64 nodes, three message passing layers $K = 3$, and choose two three-layer neural networks for $\gamma^{(k)}$ and $\phi^{(k)}$ where the hidden layers are of the size $6 \times 128$, $128 \times 128$ and $128 \times 256$ respectively. We set the learning rate to 0.001 using ADAM optimizer and a step learning scheduler of 0.2 after every 5 epochs. Note that the network architecture and hyperparameters remains the same for different simulations, unless mentioned otherwise. We find that the MSE error starts at 179.2 and reduces to $7 \times 10^{-7}$ after 20 epochs. The relative $L_2$ on the test data after training falls below $6 \times 10^{-6}$. The results for the prediction and ground truth for different boundary conditions on test simulations that the network has never seen are shown in figure 3. We further use the same generated data, however with more number of nodes (changing from 64 to 128) in the graph neural network and find that the error on the MPGNN on the test data does not change (mean of relative $L_2$ error on the test data changes from $5.2 \times 10^{-6}$ to $5.1 \times 10^{-6}$).

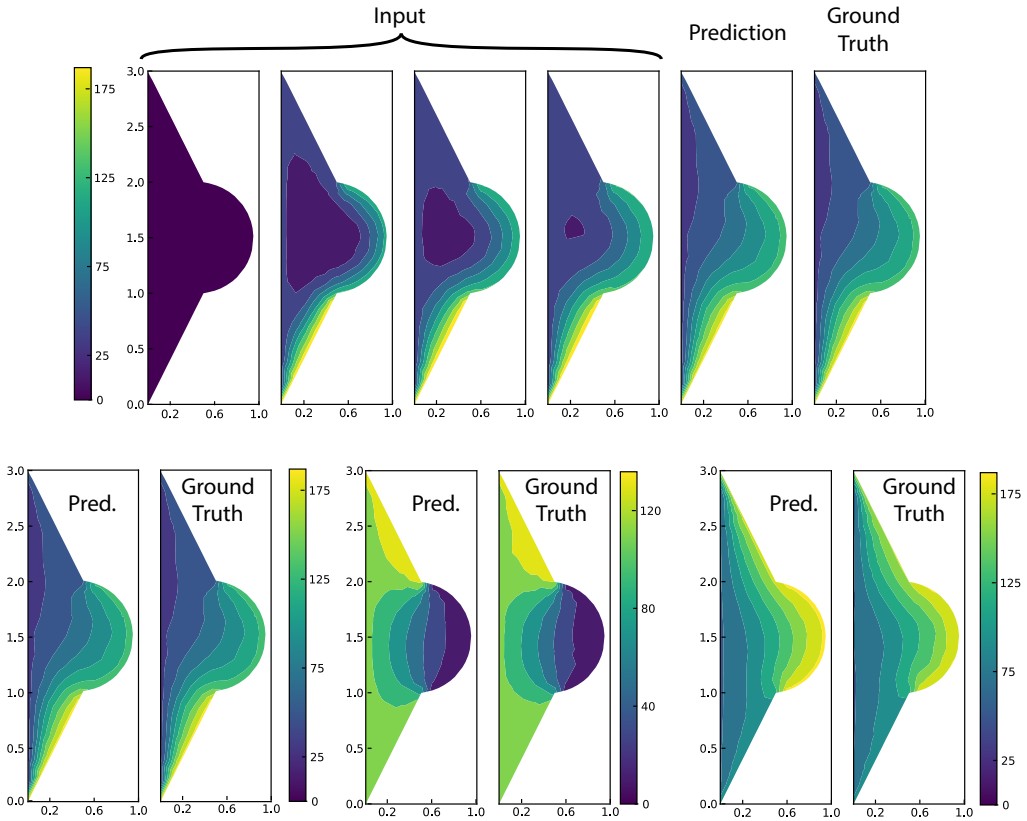

Figure 4: MPGNN trained on a square domain, predicting the values on a new unseen physical domain. We train an MPGNN to predict the heat equation on a square domain (see figure 3). Next, we use the sample MPGNN to predict the PDE values on an unseen geometry and boundary conditions.

Next, to show that the learned MPGNN can generalize to different physical domains, we create a distorted physical domain with a curved boundary, two inclined edges, and a vertical wall. The newly generated geometry is different from the initial square geometry and the network has never seen such domains. We introduce a random mesh on this new physical domain with a similar characteristic mesh size and use the MPGNN (that is only trained on the square domain) to predict the results. We find that the MPGNN indeed predicts the output with high accuracy (average MSE loss of $2.8 \times 10^{-4}$). Examples of different simulations with different boundary conditions are shown in figure 4. This shows that the graph neural network here learns to predict the future values independent of the initial physical geometry and can be extended to predict a PDE results on new unseen physical domains. Here we have used four previous snapshots as the input to our network. In order to show the effect of number of input frames on the prediction power of our network, we run tests by changing the number of input frames

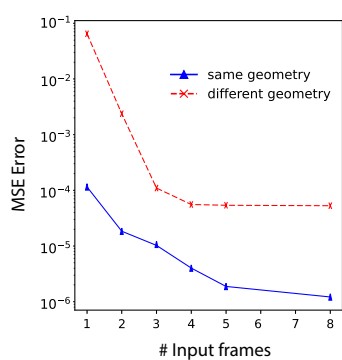

Figure 5: MSE error tried on the same and different geometry versus number of input frames

and calculating the final average MSE error on the test data set on both similar and different geometry. In all the cases we predict the frame at $80\delta t$ after the final frame. When we change the number of input frames, the input frames are respectively $40\delta t$, $25\delta t$, $20\delta t$, $15\delta t$, and $10\delta t$ apart for two, three, four, five, and eight input frames respectively. Interestingly, we find that two input frames are enough for prediction on similar geometries, however to have a reliable MSE on a different geometry we need at least three to four frames.

Lastly, we show that the MPGNN is capable of predicting sequence of temporal data, we use our MPGNN in a recurrent format (see figure 6a). We first train our MPGNN with a sequence of three past results (i.e, $n = 3$) to predict the next frame (i.e., $m = 1$). As a result, in the trained network, given input data of initial three frames of a simulation $(t_1, t_2, t_3)$, it can predict the next time step solution $t_4$. Next, we concatenate the last three obtained results, i.e. $(t_2, t_3, t_4)$ to predict the next time frame and so on. The results for a sample test case is shown in figure 6b. It is to be noted that we did not use a recurrent loss function to train our network, and as a result, the average MSE loss keeps growing with the number of predicted frames. The average MSE error for 1000 different simulations along with the range of error are shown in figure 6c where the error slowly increases with the number of predicted frames. It is expected that training the MPGNN with a recurrent loss function would improve the overall result.

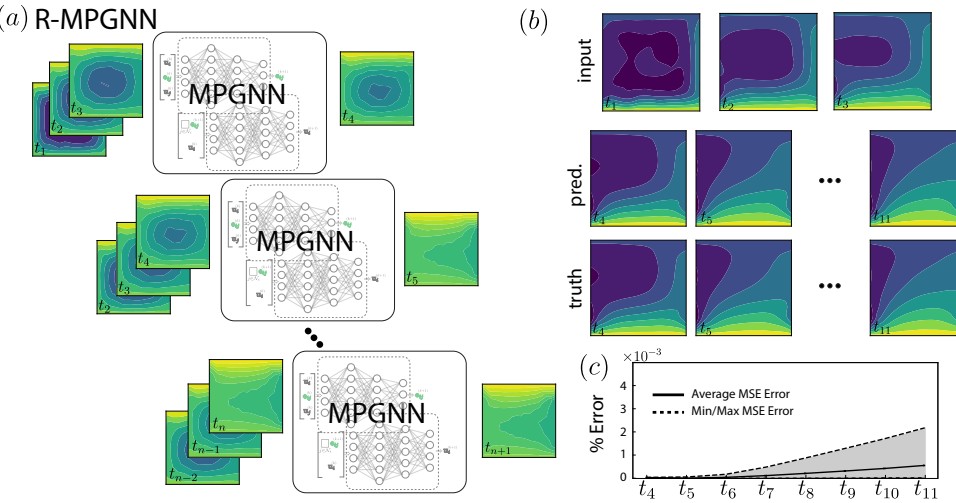

Figure 6: (a) Schematic of a recurrent message passing graph neural network (R-MPGNN) learning to predict the heat equation: input data are the initial three frames $(t_1, t_2, t_3)$ and the network predicts the next time step $(t_4)$. In the next layer the output of the previous layer $t_4$ is concatenated to the last two frames $t_2, t_3$ to create the input for the next layer $(t_2, t_3, t_4)$, and so on. (b) Test data input and outputs for the R-MPGNN predicting the next frames in time in comparison with the ground truth. (c) The average of the MSE loss for each frame prediction over time.

## 6.2 NAVIER-STOKES EQUATION

In order to find MPGNN's performance on learning nonlinear PDEs, we use a two-dimensional incompressible Navier-Stokes equation as

$$\partial \boldsymbol{v}_t + \boldsymbol{v} \cdot \nabla \boldsymbol{v} = -\frac{1}{\rho} \nabla p + \nu \nabla^2 \boldsymbol{v}$$
$$\nabla \cdot \boldsymbol{v} = 0 \tag{5}$$

where $\boldsymbol{v} = (u, v, 0)$ is the velocity field, $p$ is pressure, $\rho$ is the fluid's density, and $\nu$ is the diffusion constant. The second equation is known as the incompressibility equation which assures a divergence-free velocity field (i.e., $\nabla . \boldsymbol{v} = 0$). We use spectral methods to solve the above equation on a square domain with periodic boundary conditions (see supplementary material for the details). We chose a square domain with $[0, 2\pi] \times [0, 2\pi]$ and input a random initial condition to generate the initial conditions we sample random numbers uniformly in the range $[0, 5]$ for the nodes in the graph and then we then remove the high frequency patterns by taking a Fourier transformation and discarding the top 1/3 of high frequency wave numbers. Furthermore, in this setting our $\rho = 1$ and $\nu = $3e-4. and run our numerical PDE solver for $T = 50$ and record frames with $\delta t = 0.002$. Note that $u_i$ for different simulation also remains in the same range as $u_i \in [0, 4]$. We chose $\delta t$ such that the results become visually different. We first input five different frames (i.e., $n = 5$) and predict the next frame (i.e., $m = 1$). We generate 20000 data points from 25 simulations similar to the heat equation data

generation. In this case, we use the same network geometry for the message passing neural network as the one used for the heat equation. We find the relative $L_2$ error for the test data after 25 epochs drops from 10.1 to below $5.4 \times 10^{-5}$. The results for three different test cases are shown in figure 7. Note that in here in the plots, we are presenting the vorticity field $\zeta = \partial_x v - \partial_u u$ instead of velocity fields separately. The result here shows that the nonlinear PDE structure can also be learned using the MPGNN architecture.

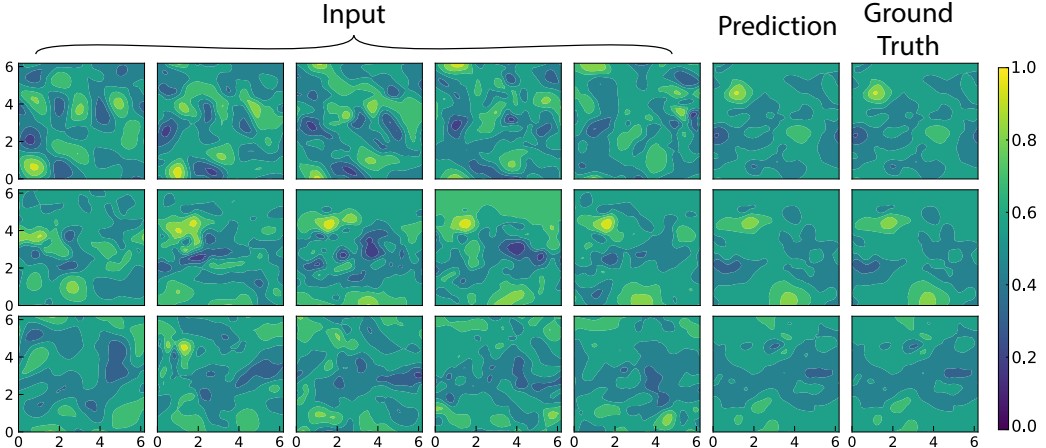

Figure 7: MPGNN trained Navier-Stokes equation with periodic boundary condition. Each row represents a different test simulation, where the first five columns are the input frames and the last two columns are respectively the MPGNN prediction and the ground truth result. We find that the average $L_2$ error for the test data is $5.4 \times 10^{-5}$.

### 6.3 ADVECTION-DIFFUSION EQUATION

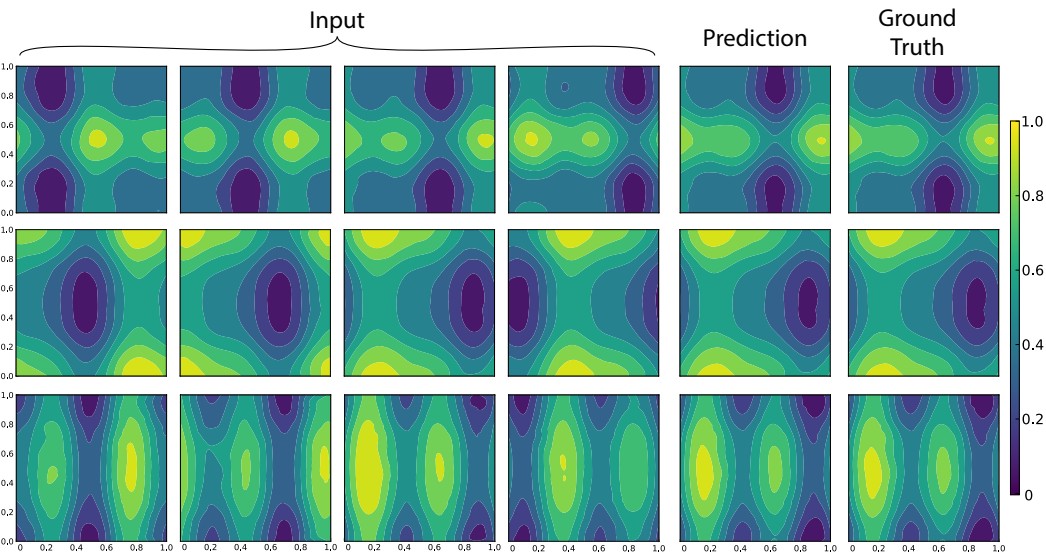

Figure 8: MPGNN trained to predict advection-diffusion equation with horizontal periodic boundary. Each row represents a different test simulation, where the first five columns are the input frames and the last two columns are respectively the MPGNN prediction and the ground truth result. We find that the average MSE loss for the test data is $1.3 \times 10^{-6}$.

In all the PDEs we tried so far, we kept the PDE parameters as constant. In this section, we focus on the advection-diffusion equation where the PDE includes different free parameters that can be used as edge features in the MPGNN. The advection-diffusion equation is

$$u_t + \lambda_1 u_x = \lambda_2 \nabla^2 u \tag{6}$$

where $\lambda_1$ and $\lambda_2$ are the advection velocity and diffusion constant respectively. We chose our domain to be $[0,1] \times [0,1]$ and use characteristic mesh size of $\delta x = 0.0625$ and time stepping of $\delta t = 10^{-4}$. We record the data every $200\delta t$ and use four subsequent input frames (i.e., $m = 4$) and predict the next frame (i.e., $n = 1$). We use `Firedrake` (Rathgeber et al., 2016) to simulate our PDE with random initial meshes and initial conditions. We use periodic boundary conditions at $x = 0, 1$ and Neumann boundary conditions at $y = 0, 1$. The initial condition is created using $u_0 = a_a \sin(x) + a_2 \sin(2x) + a_3 \cos(x) + a_4 \cos(2x)$ where all $a_i, i = 1, \cdots, 4$ are drawn from a uniform distribution $\mathcal{U}[-1, 1]$. Note that $u$ for different simulation also remains in the same range as $i \in [-3.5, 3.5]$. We select $\lambda_1$ and $\lambda_2$ randomly from uniform distributions $\mathcal{U}[0.5, 1.5]$. In total we generate 300 simulations with random initial conditions and random PDE parameters and generate 300k input data for our network. We find that in training our network the average $L_2$ error for the test data (new simulations that network has never seen) drops from 3.4 before training to $1.3 \times 10^{-6}$ after training. Three different test results are shown here in figure 8.

## 7 BENCHMARK TESTS

In this section we compare our approach with other existing relevant studies. We have identified two relevant neural network approaches closely related to our work: (1) Iakovlev et al. (2020) and (2) Li et al. (2020b) . Since we are interested in generalization of the network to other domains and mesh sizes, we perform two different tests on all of the networks and compare their performance by measuring average MSE for prediction on the test dataset. We select the heat equation as our basis (see Sec. 6.1) to test different approaches on a square geometry and test it on the unseen domain discussed in Sec. 6.1. We perform two set of tests: training on a square domain with a small characteristic mesh length (average edge size is 0.1 which we call high res), and testing the performance on the same geometry, and on a different geometry (shown in Fig. 4) with similar resolution as well as a lower resolution (average edge distance of 0.2 which we call low res). Next, we repeat the same experiment but with training on low resolution data. The results are summarized in Table 1. In repeating the work by Li et al. (2020b), we used their provided code at the corresponding git repository and used their suggested hyper-parameters and made use of $u_i$ as node features, and the position of the nodes as edge features. As seen in table 1, the proposed network has a lower MSE best when it is applied to the same geometry, however, in generalization it performs poorly which is due to the positional feature of the nodes instead of the invariant edge distance. In following Iakovlev et al. (2020), similar to the previous case, we followed the proposed structure by the authors (i.e., three hidden layers with 60 neurons along with `tanh` activation function and one message passing layer). The main difference here is that the edge features contain relative positions as the edge features instead of absolute positions. This is indeed the main reason for a better generalization of this approach compared to the other method. Our method with a different activation function (ReLU instead of `tanh`) as well as three layers of message passing, along with extra feature flagging the boundary nodes shows an improved result.

| MSE on test data | Trained on high res | | | Trained on low res | | |
|---|---|---|---|---|---|---|
| | **Our work** | (1) | (2) | **Our work** | (1) | (2) |
| same geometry (same res) | **4.01e-6** | 3.33e-5 | 1.49e-6 | **4.53e-6** | 6.27e-5 | 2.41e-6 |
| different geometry (low res) | **7.98e-5** | 1.57e-4 | 9.87e-3 | **1.57e-4** | 1.63e-4 | 1.47e1 |
| different geometry (high res) | **5.56e-5** | 9.86e-5 | 5.34e-3 | **8.06e-5** | 1.11e-4 | 5.26e0 |

Table 1: Comparison between our MPGNN, with (1) Iakovlev et al. (2020) and (2) Li et al. (2020b). We train all networks using different mesh resolution and test it on similar resolution networks, moreover to a different geometry with two different resolution.

## 8    CONCLUSION

In this paper, we showed how message passing neural networks can parametrize time-dependent partial differential equations and their discrete classical PDE solvers. Inspired by numerical PDE solvers, we introduce domain invariant features for an efficient representation of PDE data. Using graphs to represent an unstructured mesh, we trained message passing graph neural networks (MPGNN) to efficiently learn accurate solver schemes for linear/nonlinear PDEs independent of the initial trained geometry. We showed that MPGNN can predict single/multiple frame(s) of the future data and furthermore a trained model can predict results on unseen geometries. We further proposed a recurrent version of MPGNN to march in time and find a temporal sequence of solutions to a PDE. In summary, the main important features of a message passing graph neural network that make them suitable platforms for learning the time-dependent PDEs are: (i) MPGNNs similar to time-dependent PDEs are spatially locally dependent where each point is only affected by the neighboring points for a small time-step, (ii) the permutation invariant combination of the messages signifies the rotational and translational symmetries in a given physics-based PDE and helps the network to learn more efficiently, (iii) the neural networks used in creating the messages and the updates are general learners that can potentially learn the nonlinear update rules, and lastly (iv) the possibility of running several passes in message-passing neural networks helps the update of a node to see features of neighbors of neighbors and as a result MPGNN can find the update rule for larger time-steps. The possible future extension of our current MPGNN solver are: 1) including random long range connections in the initial mesh to enable long-time predictions with less number of message-passing layers; 2) training on a recurrent loss function and addressing significant memory required in backpropagation in order to improve long time predictions, ;3) using more accurate temporal schemes rather than Euler discretization to improve predictions, 4) including symbolic regression analysis to uncover the discretized kernel learned by the MPGNN and exploring the possibility of uncovering new PDE solver schemes.

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

## A    SPECTRAL METHODS SOLUTION TO NAVIER-STOKES EQUATION

Spectral methods solution to the Navier-Stokes equation. In this section, we review the spectral methods used to solve the 2d incompressible Navier-Stokes (NS) equation (5). In a 2D-fluid motion with $\mathbf{v} = (u, v, 0)$, the NS equations (equation equation 6.2) in the expanded form are

$$\partial_t u + u\partial_x u + v\partial_y u = -\partial_x p + \nu\nabla^2 u \tag{7}$$

$$\partial_t v + u\partial_x v + v\partial_y v = -\partial_y p + \nu\nabla^2 v \tag{8}$$

$$\partial_x u + \partial_y v = 0 \tag{9}$$

We define a stream function as $u = -\partial_y \psi$, $v = \partial_x \psi$. The incomprehensibility condition, equation 9, is immediately satisfied. In order to remove the pressure from the equations, we $\partial_x$ (Eq. 8) - $\partial_y$ (Eq. 7) to find

$$\partial_t \nabla^2 \psi + u\partial_x \nabla^2 \psi + v\partial_y \nabla^2 \psi = \nu\partial^4 \psi \tag{10}$$

Note that this $\nabla^2\psi = \partial_x v - \partial_y u = (\nabla \times \mathbf{v}) \cdot \hat{\mathbf{z}}$. We can simplify the last vorticity equation, equation 10, in terms of $\zeta = \nabla^2\psi$ to find

$$\partial_t\zeta + \underbrace{\left(-\partial_y\nabla^{-2}\zeta\right)}_{=u}\partial_x\zeta + \underbrace{\left(\partial_x\nabla^{-2}\zeta\right)}_{=v}\partial_y\zeta = \nu\nabla^2\zeta \tag{11}$$

Next, we discretize the equation,

$$\partial_t\zeta = \left(\partial_y\nabla^{-2}\zeta\right)\partial_x\zeta - \left(\partial_x\nabla^{-2}\zeta\right)\partial_y\zeta + \nu\nabla^2\zeta \tag{12}$$

$$\frac{\zeta^{n+1} - \zeta^n}{\delta t} = \left(\partial_y\nabla^{-2}\zeta^n\right)\partial_x\zeta^n - \left(\partial_x\nabla^{-2}\zeta^n\right)\partial_y\zeta^n + \frac{\nu}{2}\left(\nabla^2\zeta^n + \nabla^2\zeta^{n+1}\right) \tag{13}$$

where we used the implicit Crank-Nicolson for the linear part and explicit part for the linear part. Assuming periodic boundary conditions in both $x$ and $y$ direction, and taking the Fourier transform of the above equation, we find

$$\frac{\hat{\zeta}^{n+1} - \hat{\zeta}^n}{\delta t} = \widehat{\mathcal{F}(\zeta^n)} - \frac{\nu}{2}\left(|\mathbf{k}|^2\hat{\zeta}^{n+1} + |\mathbf{k}|^2\hat{\zeta}^n\right),$$

$$\mathcal{P}(\zeta) = \left(\partial_y\nabla^{-2}\zeta\right)\partial_x\zeta - \left(\partial_x\nabla^{-2}\zeta\right)\partial_y\zeta \tag{14}$$

$$\left(1 + \frac{\nu\delta t}{2}|\mathbf{k}|^2\right)\hat{\zeta}^{n+1} = \widehat{\mathcal{P}(\zeta^n)} + \left(1 - \frac{\nu\delta t}{2}|\mathbf{k}|^2\right)\hat{\zeta}^n \tag{15}$$

$$\hat{\zeta}^{n+1} = \frac{\widehat{\mathcal{P}(\zeta^n)} + \left(1 - \frac{\nu\delta t}{2}|\mathbf{k}|^2\right)\hat{\zeta}^n}{\left(1 + \frac{\nu\delta t}{2}|\mathbf{k}|^2\right)} \tag{16}$$

where $\hat{\zeta} = \mathcal{F}(\zeta)$ is the Fourier transform of the vorticity field, and $\mathcal{P} = \mathcal{F}\left(g(\zeta^n)\right)$ where $g(\zeta^n) = \left(\partial_y\nabla^{-2}\zeta^n\right)\partial_x\zeta^n - \left(\partial_x\nabla^{-2}\zeta^n\right)\partial_y\zeta^n$. We use equation 16 to propagate in time and to solve for the vorticity field. Given the vorticity field, we can further calculate the horizontal and vertical velocity with $u = -\partial_y\nabla^{-2}\zeta, v = \partial_x\nabla^{-2}\zeta$.

## B  PERIODIC BOUNDARY CONDITION

In order to impose the periodic boundary condition, we add connections to the boundary nodes that would result in a periodic boundary condition. Particularly, we consider similar copies of the nodes and by repeating the Delaunay triangulation for the nodes, we find how the boundary nodes are connected with the other the inside and boundary nodes (see Fig. 9).

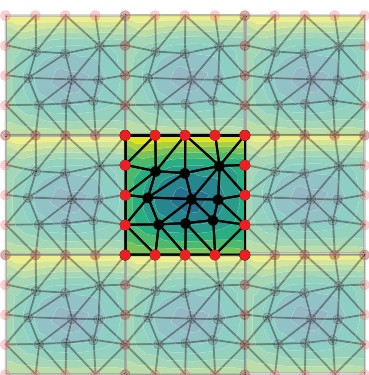

Figure 9: Schematic of a nodes and edges for a simulation with a periodic boundary condition.

