# OpenReview forum: "Learning Time-dependent PDE Solver using Message Passing Graph Neural Networks"
_ICLR.cc/2022/Conference — ICLR 2022 Submitted_

### Official Review · Reviewer_Eu7u · 2021-11-01

**Correctness:** 3
**Technical Novelty And Significance:** 2
**Empirical Novelty And Significance:** 2
**Recommendation:** 5
**Confidence:** 4

**Details Of Ethics Concerns:**

No ethics statement was submitted

**Main Review:**

Advantages

- The submission until the experimental section is well-written, easy to follow, and very clear. The written-style could include more PDE fundamentals for the uninitiated reader, but I think on the whole the authors have chosen for a clearer and simpler-to-follow read.

- The proposed GNN architecture makes a lot of sense. Using a GNN on the mesh is intuitive and the particular features used are also sensible. The particular from of edge embedding used also makes the GNN locally shift equivariant. Furthermore, encoding the PDE parameters in the edge features allows the model to generalise across PDEs within a given class of PDEs.

- The experiments demonstrate that the MPGNN can learn to make 1-step PDE prediction on multiple geometries. This is evidences in Figures 2 and 3.

Queries

- Just under equation 2: The argument that F(u, \nabla u, …) contains local spatial derivatives an only local neighbouring values and so you only need a graph neural network operating on local neighbourhoods makes sense. I was wondering, what are the limits on this assumption? Does the solution space have to be analytic/smooth? I’m guessing the maximum derivative order and accuracy of approximation will also influence the size of neighbourhoods you should take.

- Section 5: do you only use 1-step neighbourhoods in your model? I could imagine longer range message passing is important in PDEs with higher-order derivatives.

- Experiments: It is not obvious to me what is going on in your experimental setup. I would not be able to reproduce these experiments. Specifically:

— Does the MPGNN output 1 time frame at a time, but it takes in 4?

— Why are the 4 input frames separated by 20 \delta t each?

— How are the coordinates for the mesh chosen?

— From which distribution do you sample initial conditions and boundary conditions?

— Which optimiser do you use?

— Is the loss a 1-step loss?

— Do you predict values on the boundary or not?

- If you run the learned solver in recurrent mode, the error appears to increase over time as in Figure 4. c). Have you checked to see what happens over long rollouts? Does the error increase gradually or blow-up exponentially? I think it would be important to consider the long term error behaviour, since this is very important for time-dependent PDE solvers.

- Baselines: I do not see any baselines. Without these it is difficult to make sense of the number in the submission. Sensible baselines could have been finite difference method, finite volume method, spectral method, off-the-shelf software, or other neural methods, such as the Fourier Neural Operator, or the method of Bar-Sinai et al. (2019).

- What is the connection between this work and “Learning to Simulate Complex Physics with Graph Networks” (Sanchez-Gonzalez, 2020)? Here, the authors also use MPGNNs to solve complexity physical systems, driven by PDEs.

Minor notes

- Just under equation 2: n is a superscript, not subscript

- Heat equation: Please qualify what Firedrake is. Is is a software package? This is not indicated in the text.



**Summary Of The Paper:**

A message passing graph neural network is used to predict the time-dependent solution to a family of PDEs. The PDE parameters, initial conditions, and boundary conditions are input to the network, an it outputs the solution 1 time-step into the future. The authors show how to train such a system on the Heat equation, Navier-Stokes, and advection-diffusion equations. The flexibility of the message passing graph neural network permits its deployment on a diverse set of geometries and boundary conditions.


**Summary Of The Review:**

The method is well laid out and written with enough clarity that the setup is understandable and intuitive. The method makes sense given the problem and is a logical step forward in this space. The experiments, however, are insufficient to back up the claims laid forth in the first section of the paper. I do not think I could reimplement the experiments based on the writeup and I do not see a separate supplementary material, where I may have missed these details (please correct me if I have overlooked them). Furthermore there is no comparison with any baselines (I’ve mentioned a few sensible ones in the main review), neural or classical, so it is hard to say how significant the numbers in the submission are. There are also no ablations for the reader to understand the impact of each design choice the authors make. It is mainly because of the experimental section that I recommend a reject for this submission.

---

> ### Author Response · Authors · 2021-11-23
> **Response to Reviewer Eu7u**
>
> **We thank the Reviewer for carefully reading our manuscript and for their insightful comments. We have addressed all the comments raised by the reviewer, point by point.**
>
> *  **Just under equation 2 ...**
>
> Indeed the number of derivatives in the PDE affects the required number of neighboring points to be included. This can be seen from a 1d example, where for order $d$ derivative at least $N>d$ points are required in a finite difference estimation scheme. We have used a depth of 3 for our message passing, which roughly speaking allows connections to 3rd order neighbors and resolving up to 6th order derivative estimation. In all of our examples, we at most have a second-order derivative estimation. Another assumption here is that the functions are smooth so that the derivatives can be approximated using discrete points. We have included these points in the manuscript.
>
> * **do you only use 1-step neighbourhoods?**
>
> In our network we have a depth of 3, meaning that we are having three levels of message passing (we can estimate at most 6th order derivatives, however the highest spatial derivative in the paper is 2nd order).
>
> * **Does the MPGNN takes in 4?**
>
> Our MPGNN takes 4 previous frames as the input. We have also included a study on how the number of input frames would affect the MSE on the test data.
>
> * **Why are the 4 input frames separated by $20 \delta t$ each?**
>
> The main reason is to show that our model is able to take much larger timesteps than classical solvers while avoiding performance bottlenecks.
>
> * **How are the coordinates for the mesh chosen?**
>
> The position of the nodes is chosen randomly using the Poisson point process. To clarify this process we have added more information to the main text.
>
> * **From which distribution do you sample initial conditions and boundary conditions?**
>
> We sample the initial data and boundary conditions uniformly for all the test cases. In the heat equation, the initial condition is always zero, while the boundary conditions are sampled with a uniform distribution from $\mathcal{U}[0,200]$. In the Navier stokes equation, we sampled the node values using a uniform distribution $\mathcal{U}[0,5]$ and then discard the top $1/3$ of high-frequency waves. Particularly, we take the Fourier transform of the initial condition and then discard the top $1/3$ of high-frequency waves to remove the high-frequency noise in our input data. The boundary condition in the Navier-Stokes equation is periodic and requires no sampling. In the advection-diffusion equation, the initial data is sampled from $u_0 =  a_a \sin(x) + a_2 \sin(2x) + a_3 \cos(x) + a_4\cos(2x)$ where all $a_i, i=1,\cdots,4$ are drawn from a uniform distribution $\mathcal{U}[-1,1]$. We have made sure that all this information is mentioned in the main text.
>
> * **optimiser?**
>
> We have used adam optimizer for the training process. We have included the details of the algorithms used and the hyper-parameters used in the main manuscript.
>
> * **1-step loss?**
>
> The loss is indeed a 1-step loss. This is the main reason why we see the MSE loss increases with increased unrolling steps. Unrolled loss in a recurrent version requires significant memory in the backpropagation through GNN. Resolving this challenge is our current ongoing work which we aim to accomplish in our follow-up works. We have added this in our conclusion to point out the possible improvements and future directions.
>
> * **predicting values on the boundary?**
>
> The boundary values have a feature to signify that they are boundary nodes. In the presented model, the boundary node values are also predicted and are included in the MSE loss. We have tested both inclusion and exclusion of boundary nodes, and we found that including them slightly improves the results. To avoid confusion this has now been included in the main text.
>
> * **what happens over long rollouts?**
>
> The error increases exponentially over long roll-outs due to the fact that we optimize over a single step. We are currently working on recurrent loss functions, however, it becomes challenging due to significant memory requirements in backpropagation. We hope to share our ongoing work in a follow-up.
>
> * **Comparison with baselines**
>
> We have now included a section (see section 7) to compare other relevant models with our approach. Particularly, we have identified two other graph neural network approaches and compared the performance of our approach with them.
>
> * **Connection with Sanchez-Gonzalez, 2020?**
>
> The method in Sanchez-Gonzalez, 2020 is based on particle-based models of solving PDEs and is different from our mesh-based approach. We have now included a discussion over this paper in our literature review. They create a physical simulator using a particle-based method. Particularly, the state of a physical system is expressed using particles as a reduced-order model, which is then used as nodes in a graph. This is different from a mesh-based method in solving PDEs.

---

> > ### Comment · Reviewer_Eu7u · 2021-11-25
> > **Manuscript has improved**
> >
> > _General remark: Thanks to the authors for addressing many of my questions. The changes in the paper have improved it and I've moved my score up 1 notch. It's not enough for a 6 yet though, since some things are still missing (see below). Side note, the paper is now 11 pages long!_
> >
> > - Just under equation 2 ...
> >
> > _Thanks for making this change._
> >
> > - Do you only use 1-step neighbourhoods?
> >
> > _Thanks for the clarification._
> >
> > - Does the MPGNN takes in 4?
> >
> > _Thanks for the clarification._
> >
> > - Why are the 4 input frames separated by $20 \delta t$ each?
> >
> > _Thanks for the clarification. Did you run an ablation over various separation lengths and measured the effect on performance? For instance, $1 \delta t$, $2 \delta t$, $4 \delta t$, $8 \delta t$, …. This would have been highly helpful for motivation, otherwise, this 20 seems to be somewhat arbitrary._
> >
> > - How are the coordinates for the mesh chosen?
> >
> > _Thanks for clarification and amending the text._
> >
> > - From which distribution do you sample initial conditions and boundary conditions?
> >
> > _Thanks for the clarification. Out of interest, did you consider the affect of training on ICs/BCs from your given distributions and testing out-of-distribution ICs/BCs and see what happens? Not necessary to this current submission, but an interesting research direction._
> >
> > - Optimiser?
> >
> > _Thanks again._
> >
> > - 1-step loss?
> >
> > _It would be good to include this in the discussion I think._
> >
> > - Predicting values on the boundary?
> >
> > _Thanks again. Why do you think this is the case? Does it make sense to predict boundary values, if this means that you explicitly break those boundary constraints?_
> >
> > - What happens over long rollouts?
> >
> > _Please do include your negative results in the paper. Even though rollout blow up, I think it is an scientifically important observation. Please also postulate what is going on here? Do you think training on longer rollouts is going to solve this completely?_
> >
> > - Comparison with baselines
> >
> > _Thanks for including this, this was very important. I'll raise my rating 1 notch._
> >
> > - Connection with Sanchez-Gonzalez, 2020?
> >
> > _This was very useful for clarification. Thanks again._

---

> > > ### Comment · Reviewer_Eu7u · 2021-11-29
> > > **Final response**
> > >
> > > Thanks to the authors and other reviewers in this process.
> > >
> > > This is just a final remark to stay that I am sticking with my final score for this submission of 5, up from 3, after having read the other comments. I think the core of what has been presented is certainly interesting, and this is a fascinating new area of research, but I also do believe that the authors have to polish their submission somewhat before it is conference-worthy. I encourage them to keep on improving it.

---

### Official Review · Reviewer_LoDm · 2021-11-02

**Correctness:** 4
**Technical Novelty And Significance:** 3
**Empirical Novelty And Significance:** Not applicable
**Recommendation:** 6
**Confidence:** 3

**Main Review:**

Strengths

This paper introduces a locally invariant feature representation inspired by classical PDE solvers to efficiently learn a differential operator solver. The method can obtain robust high-performance models for various linear/nonlinear PDEs with arbitrary spatial discretization and arbitrary physical domains.

Weaknesses

Some detailed instructions and experiments need to be added, such as how to construct meshes, how to construct boundary conditions, and sensitivity analysis of prediction results to data sets.

Comparisons with other methods need to be added.

Some examples in irregular computational domains can be added.

I would love to hear some limitations of the proposed method.


**Summary Of The Paper:**

This paper presents a graph neural network approach by using a message-passing model to find efficient PDE solvers. It further shows that the trained solver can find a temporal sequence of PDE solutions in different domains. The proposed method is a nice idea and has been well verified. However, given the very limited scope of application in this paper, only toy environments with high-resolution numerical data sets can be used, more detailed ablation and comparison experiments may be required to ablate the model in a more refined manner.

**Summary Of The Review:**

This paper proposes a new method, which is innovative, but only shows some simple experiments and lacks detailed discussions.

---

> ### Author Response · Authors · 2021-11-23
> **Response to Reviewer LoDm**
>
> **We would like to thank the reviewer for carefully reading our manuscript and their points. In the following, we have addressed all the comments raised by the reviewer, point by point.**
>
>
> * **Some detailed instructions and experiments need to be added, such as how to construct meshes, how to construct boundary conditions, and sensitivity analysis of prediction results to data sets.**
>
> The mesh is created using Firedrake's built-in mesh generator. The details of constructing periodic boundary conditions are added to the appendix (appendix B). We have also run a sensitivity analysis to the number of input frames and included the results on page 7 (see Fig. 5).
>
> * **Comparisons with other methods need to be added.**
>
> We have now included a section (see section 7) to compare other relevant models with our approach. Particularly, we have identified two other graph neural network approaches and compared the performance of our approach with them. We used the same generated data and compared the performance of the models in predicting over the same domain. We also tested the generalization by changing the mesh-size, and also domain geometry. We find that our model outperforms both studies. The slight performance increase between our model and the Iakovlev et al. (2020) is due to the fact that we introduce a flag as a feature if the node lies on the boundary (since the boundary nodes should be treated differently from the inside nodes), and also the fact that our architecture with multiple passes allows higher-order neighbors to affect a node’s value.
>
>
> * **I would love to hear some limitations of the proposed method.**
>
> Our method uses local information to find the update to a node and this is only possible if the time-stepping between nodes is not very large. If the time-stepping between frames is large there are two possible solutions: (1) to increase the number of input frames so that a node find more information about the neighboring points by knowing the history, or (2) to increase the number of message passes so that it results in an increase of the domain of influence for a node where it can obtain information about farther neighboring nodes. Another limitation of our method is that in time marching our error rapidly increases since we are only minimizing the loss for one step. Improving this result requires including a recurrent loss function. We have included these discussions in the conclusion.

---

> > ### Comment · Reviewer_LoDm · 2021-11-29
> > **reply**
> >
> > Thank you for your reply. Perhaps adding some examples in irregular computational domains can make the paper more convincing.

---

### Official Review · Reviewer_kLFD · 2021-11-02

**Correctness:** 3
**Technical Novelty And Significance:** 1
**Empirical Novelty And Significance:** Not applicable
**Recommendation:** 3
**Confidence:** 4

**Main Review:**

**Strengths**

- The method is described fairly clearly and is easy to understand: applying a GNN to learn solutions to PDEs.

- The authors evaluate their approach on a number of test problems and show that it can generalize to unseen domains.

- The approach also extends to time-resolved simulation.

**Weaknesses**

- To me it seems that the main contributions of this work have already been demonstrated. Pfaff et al. (ICLR 2021) have already demonstrated a GNN for solving a wide array of simulation and PDE problems. The architecture of Pfaff et al. is generally similar to the proposed approach, they demonstrate generalization to unseen domains, and the experimental results go beyond that what is shown in the proposed method (including 3D, adaptive remeshing, and dynamic simulation domains). The authors should cite this work and explain if or how their contributions differ.

Pfaff, Tobias, et al. "Learning mesh-based simulation with graph networks." In Proc. ICLR 2021.

- The proposed recurrent approach to solving time-dependent PDEs is not trained with a recurrent loss. Thus the authors note that the MSE increases with increased unrolling steps. This is an open challenge for learned PDE solvers. Usually it is difficult to incorporate an unrolled loss because of the significant memory required to backpropagate through the GNN for multiple time steps. Is it possible to train the proposed method with a recurrent loss, or is that computationally prohibitive?

**Misc.**

- The authors note that there is a periodic boundary condition for the Navier-Stokes equations they solve. Do they incorporate this into the mesh they use in any way (e.g., using a periodic mesh to help the network learn these boundary conditions)?


**Summary Of The Paper:**

The paper presents a method for learning mesh-based simulation of partial differential equations using graph neural networks (GNNs). The authors train the GNN to match a reference solution and demonstrate that the trained GNN can generalize to solving PDEs on different domains. Additionally, the authors show a recurrent variant of the network that can solve time-dependent PDEs. The method is shown to work on a number of test problems, including advection-diffusion and Navier-Stokes equations. Overall, the authors show that GNN-based simulation is a promising approach for improving the flexibility and generalization of learned PDE solvers.


**Summary Of The Review:**

While the paper seems generally well written and the method appears sound and thoroughly evaluated, I'm not sure what the novelty of the paper is given that the contributions overlap significantly with Pfaff et al. (2021). Moreover, the authors do not cite this paper or explain how their work differs from it. Thus, my rating is to reject the paper.

---

> ### Author Response · Authors · 2021-11-23
> **Response to Reviewer kLFD**
>
> **We would like to thank the reviewer for carefully reading our manuscript and their points. In the following, we have addressed all the comments raised by the reviewer, point by point.**
>
> * **The authors should cite this work [Pfaff, Tobias, et al.] and explain if or how their contributions differ**
>
> We thank the reviewer for bringing this manuscript. Since it is very relevant to our study, we have included this in our literature review. Indeed both our work and Pfaff, Tobias, et al. show the relevance of using MPGNNs in physical simulations and proving the generalization and extensibility. There are some technical differences between our work and Pfaff et al. The main difference is that Pfaff et al. encode the data with an MLP in a graph and then process the data in the encoded domain rather than the physical domain. We, however, remain in the physical domain to increase the interpretability of the messages introduced for each mesh node. The second difference is that Pfaff et al. find the derivative updates and use a time integrator to find the new values, while we are directly predicting the next value and the integrator is encoded in the graph network. The reasoning behind our work is in Eq. 3 being a far more general form of Eq. 2. We have now included a finite difference example on page 3 to show the motivation behind our work and how our approach is rooted in classical PDE solver schemes. We have also now included a discussion on Pfaff et al. in our literature review.
>
> * **Is it possible to train the proposed method with a recurrent loss, or is that computationally prohibitive?**
>
> This is indeed our current ongoing work that we are aiming to accomplish in our follow-up work. As the reviewer points out this is an open challenge due to the significant memory required in backpropagation. We have added this discussion in our conclusion.
>
> * **How is the periodic boundary condition implemented?**
>
> We have now added an extra section to the appendix discussing the process of a periodic boundary condition. In order to impose the periodic boundary condition, we add extra edges to connect nodes on the boundary such that it would result in a periodic boundary condition. To achieve this goal, we consider similar copies of the nodes, and then by repeating the Delaunay triangulation for the nodes, we find how the boundary nodes are connected with each other and the inside nodes. We then use those connections to construct our computational graph. Please see the Periodic Boundary Condition section On page 14  of the appendix (appendix B) for more information.

---

> > ### Comment · Reviewer_kLFD · 2021-11-27
> > **response**
> >
> > I'd like to thank the authors for their response to my review. I also appreciate the clarification with respect to the related work of Pfaff et al.
> >
> > While there are some technical differences between the approaches, it's not clear if the differences in the proposed method actually provide any quantitative improvement over Pfaff et al. The overall approach seems to be very similiar, and the results of Pfaff et al. already seem to go beyond what is proposed (e.g., including 3D and dynamic simulations). As a result I would still keep my negative rating.

---

### Official Review · Reviewer_neAM · 2021-11-05

**Correctness:** 3
**Technical Novelty And Significance:** 2
**Empirical Novelty And Significance:** 1
**Recommendation:** 5
**Confidence:** 5

**Main Review:**

The paper is well organized with intuitive explanations of the network design approach and detailed experiments. The strength of the paper is on adopting graph neural networks. However, based on my understanding, there are a few weaknesses (and questions that could help reviewers understand the paper better).

1. As mentioned by the authors, the domain invariant feature representation is the major contribution and is inspired by PDE solvers. However, except for highlighting the approximation by using the neighboring points in the domain, it either lacks more in-depth analysis or lacks novelty as, for example, using finite difference etc, in a neural network for PDEs is a not unknown approach. Could you be more specific with more details on the novelty?

2. Section 5 introduces the proposed network with nice illustrations. Is it just a typical architecture of message passing neural network with some fully connected layers. Could you please be more specific on the novelty on the structure? As authors mention in section 6.1 that different domains were tested to show the generalization capability, could you be more specific on which part of the network that would allow good generalization to be domain invariant?

3. The experiment section is well written with detailed explanation and visualization. However, there are a few questions:

3.1 What is the scale of the data, i.e. $u_i$? Without reference, I find it is hard to measure the claimed MSE that was achieved.

3.2 What is the training algorithm, or in other words, the optimizer chosen for training the network?

3.3 What is the baseline that the proposed method is comparing with? How about other networks that was mentioned in the related works? Lacking these two important reference makes it hard to measure the performance of the proposed method.

3.4 While the result visualization is nicely presented, without detailed numerical presentation, I am not sure how well the performance is and how efficient the training is?

Some missing references:

[1] Physics Informed Deep Learning (Part II): Data-driven Discovery of Nonlinear Partial Differential Equations

[2] Solving high-dimensional partial differential equations using deep learning

[3] Finite Difference Neural Networks: Fast Prediction of Partial Differential Equations

**Summary Of The Paper:**

This paper proposes to use a message passing neural network to predict solutions of both linear and nonlinear PDEs. The proposed method uses domain invariant features with graph based method for representing the temporal data of PDE.

**Summary Of The Review:**

To summarize, this paper basically applies message passing neural networks to solving PDEs. I strongly recommend authors to present motivations and design of networks with more details and analysis, and the experiment requires improvement.

---

> ### Author Response · Authors · 2021-11-23
> **Response to Reviewer neAM**
>
> **We would like to thank the reviewer for carefully reading our manuscript and their points. In the following, we have addressed all the comments raised by the reviewer, point by point.**
>
> * **Could you be more specific with more details on the novelty?**
>
> The method proposed here is inspired by the finite difference method and the possibility of approximating a partial differential equation using discrete point stencils. The finite difference method is usually derived on structured meshes and generalizing the derivative approximations for a general positioning of the nodes is a computationally expensive task and usually, results in unstable methods. Other methods such as Finite Elements Methods or Finite Volume Methods are used instead in more general cases. Here, we use graph neural networks as a nonlinear function approximator and use the simulated data to learn the required stable stencils. Our approach here has a physical basis, and we show that the kernel solver learned is generalizable as it can find the PDE solutions (both linear and nonlinear) in different geometries (both on initially trained geometries and new unseen domains). The other main contribution here is the use of our graph neural network in a recurrent approach to march in time and find the time-sequence solutions to a PDE. We have now included the following text in the main text to make our contributions more clear (see page 3).
>
> * **Could you be more specific on which part of the network would allow good generalization to be domain invariant?**
>
> The update rule in message-passing graph neural network with the carefully selected features (i.e., Eq. 3) is capable of learning an update rule for a time-dependent PDE (i.e., Eq. 2) in its general form. To make this point concrete, we now provide an example of Eq. 2 for the heat equation and show how this can be seen using the update rule in message-passing neural networks. In summary, the main important features in a message-passing graph neural network that makes it a nice platform for learning the time-dependent PDEs are: (i) message passing networks similar to PDEs are locally dependent on features where neighboring points are only important to find the updates, (ii) the permutation invariant combination of the features signifies the rotational and translational symmetries of a given physics-based PDE, (iii) the generality of neural networks in creating the messages and the update rule in capturing the nonlinear update rules, and lastly (iv) the possibility of running several passes in message-passing neural networks that helps the update rule to see features of a PDE from points at a farther distance. To make this point more clear, we have added the example of a finite difference on page 3 and added discussion on page 11.
>
>
> *  **What is the scale of the data, i.e. $u_i$?**
>
> In the heat equation, $u_i \in [0, 200]$, while in the Navier-Stokes equation $u_i \in [0,4]$. In the case of advection-diffusion equation, $u_i \in [-3.5, +3.5]$. We have now included these details in the main text (on pages 5, 8, and 9)
>
> * **Without reference, I find it is hard to measure the claimed MSE that was achieved.  What is the baseline that the proposed method is comparing with?**
>
> We have now included a section (see section 7) to compare other relevant models with our approach. Particularly, we have identified two other graph neural network approaches and compared the performance of our approach with them. We used the same generated data and compared the performance of the models in predicting over the same domain. We also tested the generalization by changing the mesh-size, and also domain geometry. We find that our model outperforms both studies. The slight performance increase between our model and the Iakovlev et al. (2020) is due to the fact that we introduce a flag as a feature if the node lies on the boundary (since the boundary nodes should be treated differently from the inside nodes), and also the fact that our architecture with multiple passes allows higher-order neighbors to affect a node’s value.
>
>
> * **What is the training algorithm, or in other words, the optimizer chosen for training the network?**
>
> We have used Adam optimizer for the training process. We have included the details of the algorithms used and the hyper-parameters used in the main manuscript.
>
>
>
> We thank the referee for bringing up missing papers to our attention. We have now included all of them in our literature review.

---

> > ### Comment · Reviewer_neAM · 2021-11-29
> > **Response to authors**
> >
> > Thank you for your detailed explanations and revision. I can see the efforts in the rebuttal and updated paper did improve the paper, and that is also praised by other reviewers. But I am not convinced that the improvement is sufficient for me to adjust the initial score.  I really appreciate the efforts authors put in within such a short amount of time window. I'd like to recommend authors keep improving the experiments as, even without very obvious novelty over the existing approach/architecture, if a more extensive experiment can be performed and show a good empirical gain of the proposed method, it would still be considered as a good paper.

---

### Decision · Program_Chairs · 2022-01-20

**Decision:**

Reject

**Comment:**

Thank you for your submission to ICLR.  While all reviewers felt that there were some interesting aspects to the proposed work, the consensus was also that the work didn't properly situate itself within the existing literature on related methods.  In particular, I agree with Reviewer kLFD that a numerical comparison to Pfaff et al., is notably missing here; while the authors did provide qualitative comparisons in their discussion, it's not clear to me that these differences are ultimately that significant, and the methods need to be compared directly if a case is to be made for the advantages of the proposed approach.